# Embedded Human Periodontal Ligament Stem Cells Spheroids Enhance Cementogenic Differentiation via Plasminogen Activator Inhibitor 1

**DOI:** 10.3390/ijms23042340

**Published:** 2022-02-20

**Authors:** Madoka Yasunaga, Hiroyuki Ishikawa, Sachio Tamaoki, Hidefumi Maeda, Jun Ohno

**Affiliations:** 1Section of Orthodontics, Department of Oral Growth and Development, Fukuoka Dental College, Fukuoka 814-0193, Japan; madoka@college.fdcnet.ac.jp (M.Y.); tama@college.fdcnet.ac.jp (S.T.); 2Professor Emeritus, Fukuoka Dental College, Fukuoka 814-0193, Japan; ishikawa.hiroy@gmail.com; 3Division of Oral Rehabilitation, Department of Endodontology and Operative Dentistry, Faculty of Dental Science, Kyushu University, Fukuoka 812-8582, Japan; hide@dent.kyushu-u.ac.jp; 4Oral Medicine Research Center, Fukuoka Dental College, Fukuoka 814-0193, Japan

**Keywords:** human periodontal ligament stem cells, spheroids, collagen embedding culture, plasminogen activator inhibitor 1, cementogenic differentiation, cementum protein 1

## Abstract

Spheroids reproduce the tissue structure that is found in vivo more accurately than classic two-dimensional (2D) monolayer cultures. We cultured human periodontal ligament stem cells (HPLSCs) as spheroids that were embedded in collagen gel to examine whether their cementogenic differentiation could be enhanced by treatment with recombinant human plasminogen activator inhibitor-1 (rhPAI-1). The upregulated expression of cementum protein 1 (CEMP1) and cementum attachment protein (CAP), established cementoblast markers, was observed in the 2D monolayer HPLSCs that were treated with rhPAI-1 for 3 weeks compared with that in the control and osteogenic-induction medium groups. In the embedded HPLSC spheroids, rhPAI-1 treatment induced interplay between the spheroids and collagenous extracellular matrix (ECM), indicating that disaggregated HPLSCs migrated and spread into the surrounding ECM 72 h after three-dimensional (3D) culture. Western blot and immunocytochemistry analyses showed that the CEMP1 expression levels were significantly upregulated in the rhPAI-1-treated embedded HPLSC spheroids compared with all the 2D monolayer HPLSCs groups and the 3D spheroid groups. Therefore, 3D collagen-embedded spheroid culture in combination with rhPAI-1 treatment may be useful for facilitating cementogenic differentiation of HPLSCs.

## 1. Introduction

The periodontium supports and houses the dentition as a periodontal complex that consists of tooth root cementum, periodontal ligament (PDL), and alveolar bone [1]. Cementum is the outermost layer of the hard avascular connective tissue that covers and protects the tooth root and provides attachment sites for PDL fibers to support the tooth [2,3]. Thus, cementum formation is critical for the development of the periodontium and the regeneration of periodontal tissues; therefore, regeneration of the cementum would be highly beneficial to periodontal tissue repair. Calcified cementum is produced by cementoblasts (known as the progenitors of the cementum), which deposit a mineralized matrix onto the root surface [4].

Mesenchymal stem/stromal cells (MSCs) are multipotent somatic stem cells that can differentiate into mesodermal cells, such as osteoblasts, chondrocytes, myocytes, and adipocytes [5,6]. Previous studies have examined the use of dental mesenchymal stem cells for periodontal tissue formation [7,8]. Among the tooth-derived stem cells, human PDL stem cells (HPLSCs) express various stromal cell markers, such as STRO-1, CD13, CD29, CD44, CD59, CD90, CD146, and CD105, that have similar morphological, phenotypic, and proliferation characteristics to adult MSCs [9,10]. Adult MSCs are derived from adult human tissues and provide a potentially powerful candidate for cell-based tissue engineering and regeneration [11]. Particularly, HPLSCs have the potential to differentiate into osteoblasts and adipocytes [12,13]. Therefore, HPLSCs are considered the most promising source of stem cells for periodontal regenerative therapies, including cementum formation.

Despite the important role of cementum in regeneration after periodontal disease, the properties of its progenitors and the role of cementoblasts remain unclear. Recent studies have reported that cementum protein 1 (CEMP1; also called CP23) and cementum attachment protein (CAP) have been found to be expressed in cementogenic cells. Both proteins have been identified in the progenitor cells of the PDL and cementoblasts [10,14,15]. CEMP1, a non-collagen protein, was isolated and characterized as a human cementoblast-derived protein [14] and CEMP1 mRNA was shown to be expressed in the cementoblast cell layer lining the cementogenic surface of cementum using in situ hybridization [15]. Furthermore, immunohistochemical staining of human periodontal tissue revealed that the CEMP1 protein was localized to cementoblast cells and restricted subpopulations (cementoblast precursors) of PDL cells in the PDL space. CEMP1 is a reliable marker of cementoblasts and regulate cementoblast differentiation in the PDL space to promote octacalcium phosphate crystal nucleation during the biomineralization process, because it is highly expressed by cementoblasts and subpopulations of the PDL that are located in the paravascular zone in the periodontium [15]. CAP, as a collagen-like protein, is restrictedly expressed to periodontal cell lineages [16]. CAP is localized in the cementum matrix and PDL as well as perivascular cells of alveolar bone [14]. A recent study suggested that plasminogen activator-inhibitor type 1 (PAI-1) is an effective inducer of cementogenesis, indicating that recombinant human PAI-1 (rhPAI-1) could promote the generation of cementum-like tissue with PDL fibers into newly formed cementum-like tissue in human periodontal ligament stem cells [8,17]. Moreover, PAI-1 transfected-conditioned media can promote osteogenesis/cementogenesis in HPLSCs and human periodical follicular stem cells [17]. PAI-1 can inhibit both tissue and urokinase-plasminogen activators and plays an important role in several physiological processes, such as fibrotic disorders, metabolic disorders, and cancer [18,19,20,21]. PAI-1 is also involved in extracellular matrix (ECM) remodeling [22] and is a known regulator of bone remodeling [23]. Thus, PAI-1 is expected to contribute to the formation of hard tissue, including cementum.

A two-dimensional (2D) culture of artificially polarized cells as a monolayer is a routinely used as a cell culture technique; however, it produces various undesirable effects, such as loss of self-renewal, replication, colony-forming, and differentiation capacities, in MSCs [24,25]. Thus, three-dimensional (3D) in vitro cell culture models are becoming more widely used as a translational tool to study the behavior and differentiation response of MSCs [26]. Such 3D systems promote greater in vivo-like cell behavior compared with 2D culture as they can recreate more of the characteristic traits of the native tissue environment [27,28]. In 3D systems, multicellular aggregates of cells, termed spheroids, more accurately reproduce the tissue macrostructure that is found in vivo compared with the 2D monolayer. The limitation of MSC-based tissue engineering, such as the low reproducibility and standardization of MSC, has been postulated by the heterogeneity of the different sources of MSCs, although MSC spheroids may have great relevant secretory activity [29]. Techniques to improve MSC characterization are needed to better measure the discrepancies. In the periodontal tissue, HPLSCs, a primary contender for generating cementoblasts, are embedded in the PDL fibrils. These cells can interact with collagenous ECM to maintain their properties and to prepare for differentiation into fibroblasts, osteoblasts, and cementoblasts. Embedding spheroids within a hydrogel promotes cellular differentiation and cell–matrix attachment, making spheroids appropriate models of the HPLSC microenvironment [30,31]. Thus, embedded HPLSC spheroids are presumed to be essential to reproduce the stem cell environment of the periodontal tissues. We hypothesized that a multicellular spheroid that was contained in an ECM-derived matrix can promote cementogenic differentiation of HPLSCs in contrast to that which was observed in a 2D monolayer system. Thus, the present study examined the effects of rhPAI-1 treatment on enhanced cementogenesis in collagen-embedded HPLSC spheroids.

## 2. Results

### 2.1. rhPAI-1 Enhances Cementogenic Differentiation of HPLSCs in 2D Culture 

Since PAI-1 is known to stimulate cementogenesis, we first examined whether PAI-1 promoted cementogenic differentiation in HPLSCs in 2D culture. HPLSCs were cultured in DMEM alone (control) or in OIM without or with rhPAI-1 (100 ng/mL) for 3 weeks as previously described [8]. The protein levels of the cementogenic markers were determined using Western blot analysis (Figure 1A,B). After 3 weeks, the expression levels of CEMP1 and CAP in the HPLSCs were significantly higher in the PAI-1-treated cells than in the control (*p* < 0.01) and in the cells in OIM alone (*p* < 0.01) (Figure 1B). In the cells that were treated with PAI-1 alone, the expression levels of both CEMP1 and CAP were identical to those in the control (data not shown). Immunocytochemistry analysis revealed that CEMP1 is expressed in the nuclei of the HPLSCs that were treated with PAI-1 (Figure 1C). In the immunocytochemical-staining of CAP, the HPLSCs that were treated with PAI-1 fluoresced in the perinuclear area (Figure 1C). The percentage of the CEMP1-positive cells was significantly different between the PAI-1 (82.6% ± 10.7%) and the control (33.8% ± 5.8%; *p* < 0.001) and the OIM (33.3% ± 5.0%; *p* < 0.05) groups (Figure 1D). In the CAP staining, the percentage of positive cells was significantly higher in the PAI-1-treated HPLSCs (84.4% ± 4.4%) than in the control (31.7% ± 8.7%; *p* < 0.001) or the HPLSCs that were treated with OIM alone (31.5% ± 0.5%; *p* < 0.001) (Figure 1D). The immunocytochemistry results of the cells that were treated with PAI-1 alone were similar to those of the control (data not shown). Next, we examined ALP levels and ARS intensity in the HPLSCs that were treated with OIM or with PAI-1 for 3 weeks (Figure 1E,F). The ALP-staining intensity of HPLSCs increased in the PAI-1-treated cells compared with the OIM-treated cells (Figure 1F). Similarly, ARS staining revealed higher calcium deposition in the PAI-1 group (Figure 1F). These findings revealed that rhPAI-1 could induce differentiation of the HPLSCs into cementoblasts as mineral-forming cells in 2D culture.

### 2.2. rhPAI-1 Induces Cementoblastic Spheroids Embedded in Collagen Gel

The embedded HPLSC spheroids were established by embedding 25 spheroids into collagen gel after generating HPLSC spheroids that were cultured in low-binding plates for 24 h. Figure 2A shows phase-contrast images of the HPLSC spheroids at 0, 48, and 72 h after the embedded culture with DMEM. All the spheroids revealed a ball-like form with distinct boundaries and a round and compact shape. The size consistency was evaluated by measuring the diameter of the HPLSC spheroids between 0 and 72 h after the embedded culture with DMEM (Figure 2B). We also investigated the changes in the cell viability of the HPLSC spheroids at 0, 48, and 72 h after the embedded culture with DMEM. As shown in Figure 2C, there was no significant difference in the cell viability between the time points.

Next, we examined the effect of PAI-1 treatment on CEMP1 expression in the HPLSC spheroids that were treated with DMEM (control), with OIM alone (OIM), and with OIM containing PAI-1 (PAI-1) for 2 days. The cells in the DMEM and the OIM-treated spheroids showed lower levels of CEMP1 expression compared with PAI-1-treated spheroids (Figure 2D). There were significant differences in the percentage of CEMP1-positive cells between the PAI-1 and the control and the OIM groups (PAI-1 vs. control and OIM; *p* = 0.00123) (Figure 2E). These findings suggest that PAI-1 treatment can enhance cementogenic differentiation in the HPLSC spheroids that were embedded in collagen gel and can consequently generate embedded cementoblastic spheroids after 3D culture for 2 days.

### 2.3. rhPAI-1 Promotes Interplay between HPLSC Spheroids and Collagenous ECM

We examined the morphological changes in the embedded spheroids that were cultured in DMEM (control), OIM alone (OIM), and OIM containing rhPIA-1 (PIA-1) to elucidate the dynamics of HPSLC spheroids within collagen gel. Ball-like shapes were obtained in all the HPLSC spheroids that were embedded in collagen gel at 24 h, regardless of the treatment. The surface of embedded spheroids that were treated with PAI-1 were covered with many spines at 48 h, showing a sea urchin-like morphology (Figure 3A), whereas other groups (control and OIM) remained as ball-like spheroids. These results indicate that PAI-1 can promote cell growth from the embedded spheroids. After 72 h of PAI-1 treatment, many spheroids were disaggregated, and the isolated cells had migrated and spread into the collagen gel (Figure 3A). By contrast, spheroids in the control and OIM groups remained as ball-like or sea urchin-like shapes at 72 h (Figure 3B). To test the potential effects of PAI-1 on the motility or migration activity of the constituent cells in the embedded spheroids to collagen gels, we measured the extended area of the spheroids at 24, 48, and 72 h after embedded culture. There were no significant differences between the spheroid ranges of the groups at 24 h (Figure 3C). After 48 h, the expanded area of spheroids that were treated with PAI-1 was greater than the control and OIM-treated groups (Figure 3C). Together, these results (Figure 3B,C) demonstrate a key role for PAI-1 in promoting cell migration to the ECM in HPLSC spheroids. During cell migration from the embedded spheroids, the spheroids disaggregate (Figure 3A,B). Next, we examined the effect of PAI-1 treatment on the disaggregation of the spheroids by calculating the percentage of disaggregated spheroids using the WinROOF image analysis software. At 24 h, there was no significant difference between the disaggregation percentages of the three groups (control vs. PAI-1, *p* = 0.608; OIM vs. PAI-1, *p* = 0.966) (Figure 3D). The PIA-1 group showed a higher percentage of embedded spheroid disaggregation (93.7% ± 4.2%) at 72 h compared with the control (5.8% ± 2.3%) and OIM (16.0% ± 4.6%) groups (*p* < 0.001 for both) (Figure 3D). These findings suggest that PAI-1 promotes interplay between the spheroids and the collagen, resulting in the disaggregation of spheroids via cell migration. 

### 2.4. CEMP1 Expression Is Elevated in Embedded HPLSC Spheroids Treated with rhPAI-1

To investigate whether the interaction between outgrowth from the spheroids and collagen fiber via disaggregation of embedded spheroids that were treated with rhPAI-1 was related to enhance cementogenic differentiation in HPLSCs, we compared CEMP1 expression in 2D monolayer HPLSCs and embedded HPLSC spheroids following treatment with DMEM (control), OIM alone (OIM), and OIM containing rhPAI-1 (PAI-1) up to 14 days. At 7 days after treatment, Western blot analysis revealed significantly upregulated expression of CEMP1 in the PAI-1-treated embedded spheroids compared with the 2D monolayer HPLSC control (*p* = 0.00083), OIM (*p* = 0.01639), and 2D PAI-1 (*p* = 0.00511) groups (Figure 4A,B). Comparisons between the embedded spheroid groups revealed that spheroids that were treated with PAI-1 have higher levels of CEMP1 expression compared with control (*p* = 0.00452) and OIM-treated spheroids (*p* = 0.01328) (Figure 4A,B). Significant upregulation of CEMP1 expression was maintained in the PAI-1-treated embedded spheroids for 14 days, compared with the 2D control and other embedded spheroid groups [vs. 2D control (*p* = 0.00072); vs. 2D OIM (*p* = 0.00084); vs. 2D PAI-1 (*p* = 0.00511); vs. embedded control (*p* = 0.00023); vs. embedded OIM (*p* = 0.00031)] (Figure 4A,B). Immunofluorescence staining showed a significant increase in the percentage of CEMP1-positive cells (fluoresced nuclei) in the embedded HPLSC spheroids that were treated with PAI-1 (65.4% ± 5.0%) for 7 days compared with the control 2D culture (5.4% ± 3.0%, *p* < 0.001), OIM (20.4% ± 2.1%, *p* = 0.00212), and PAI-1 (44.2% ± 7.0%, *p* = 0.00742) groups. In the embedded spheroid groups, the percentage of positive cells was significantly higher in the PAI-1 group than in the control (13.2% ± 4.6%, *p* < 0.001) and in the OIM-treated spheroids (26.3% ± 2.8%, *p* = 0.00313) (Figure 4B,C). At 14 days after treatment, the percentage of CEMP1-positive cells was higher in the PAI-1-treated embedded spheroids (83.9% ± 5.4%) than in all the other groups (*p* < 0.01) (Figure 4C,D). These findings suggest that PAI-1 induces interplay between the disaggregated spheroids and that collagen fibers promote cementogenic differentiation in HPLSCs.

## 3. Discussion

It was previously reported that rhPAI-1 can induce the cementogenic differentiation in the periodontal ligament stem cells [8]. The present study provides three lines of supportive evidence concerning the effects of rhPAI-1 treatment on the acceleration of the cementogenic differentiation in HPLSCs. First, rhPAI-1 enhanced cementogenic differentiation of the HPLSCs in 2D culture. Second, the 3D-embedded HPLSC spheroids that were treated with rhPAI-1 could be converted to cementoblastic spheroids via the interplay between the HPLSC spheroids and collagenous ECM. Third, the embedded HPLSC spheroids that were treated with rhPAI-1 significantly promoted cementogenic differentiation compared with the 2D monolayered HPLSCs.

The present study revealed that treatment with rhPAI-1 enhanced CEMP1 and CAP expression in 2D monolayer HPLSCs, whereas the stimulation with OIM alone suppressed the expression of both. A recent study also reported that osteogenic differentiation medium downregulated CEMP1 and CAP expression in periodontal ligament stem cells [10]. These findings may indicate that the osteogenic stimulus may induce HPLSCs to differentiate only into osteoblasts while suppressing cementogenic pathways. Our results that showed the upregulation of CEMP1 and CAP by treatment with PAI-1 support the findings of other reports that examined the effects of PAI-1 on induced cementogeniesis in human periodontal ligament stem cells. Several studies have reported that CEMP1 protein expression is localized to the cementoblast cell layer lining of the cementogenesis surface of cementum [15,32]. Monolayer HPLSCs in the PAI-1-treated group were more susceptible to cementogenic differentiation than HPLSCs in the other groups. Furthermore, CEMP1 may play an important role in cementogenesis in response to rhPAI-1. A previous study examining the role of PAI-1 in osteogenesis revealed that protein levels of the osteoblastic-specific factors, Runx2 and OSX, were increased by rhPAI-1 treatment [33]. Although it remains unclear which transcription factors may be crucial for cementogenic differentiation that is induced by rhPAI-1, our findings suggest that rhPAI-1 promotes cementoblastic differentiation in 2D monolayer HPLSCs.

Collagen-embedded spheroids consist of a collagen-based ECM structure that surrounds the spheroids. Treatment with rhPAI-1 showed disaggregation of the embedded HPLSC spheroids in the early stages compared with the control and OIM-treated groups. Collagen is the most prevalent embedding material and is a component of the ECM that is unregulated within the tissue microenvironment [34]. Spheroids that lack the surrounding ECM are composed of a metabolically active outer layer and a necrotic core. Recent studies have shown that embedded cancer spheroids possess two populations: a migratory or invasive population and a nonmigratory core [35,36,37,38]. This duality in cell behavior may imply that cell–matrix interactions dominate the migratory population and cell–cell contacts dominate the nonmigratory core. In the present study, some cells in the HPLSC spheroids showed CEMP1 expression within two days after treatment with rhPAI-1, suggesting movement of HPLSC spheroids to cementoblast-like spheroids. These cementoblast-like cells within the spheroids appeared to obtain the potential for outgrowth and migration to interact with the surrounding collagen in the ECM. Our findings concerning the early disaggregation of rhPAI-1-treated spheroids may indicate that, in addition to the migratory cells that were localized at periphery portion of the spheroids, the cells that were involved in the nonmigratory phase also changed to migratory cells. Therefore, rhPAI-1 treatment may enable the interplay of embedded spheroids and collagen in the ECM, which contributes to the induction of cementogenic differentiation of HPLSC spheroids.

Enhancement of cementogenic differentiation is attributed to the interplay between collagen in the ECM and migrated cells from the embedded spheroids that were treated with rhPAI-1. In our Western blotting and immunocytochemistry findings, CEMP1 expression was significantly unregulated in the embedded spheroids that were treated with rhPAI-1 compared with all groups in the 2D culture and the control and the OIM-treated spheroid-treated groups. These findings indicate that PAI-1-treated spheroids show a greater cementogenic differentiation capacity than the other groups. The interaction between the spheroids and the ECM via integrins and substrate mechanics plays crucial roles in the enhancement of cell differentiation and survival [39]. Since the 2D monolayer models lack an ECM and cannot reflect or predict in vivo performance, 3D embedding spheroid models more accurately represent the native periodontal environment. Therefore, in spheroids that are embedded in the collagen gels, a suitable matrix may facilitate cell ingrowth and interaction with differentiating cells, as observed by the in vivo interaction between the PDLs and periodontal stem cells. In our findings of cementogenic differentiation among the embedded spheroids that were treated with various factors, such as rhPAI-1, was significantly suitable for leading the enhancement of cementogenic differentiation in the HPLSC spheroids, suggesting that rhPAI-1 treatment may be effective in mimicking the in vivo conditions between the stem cells and collagen in the ECM of the periodontal tissue. These speculations are supported by the findings of a recent study that showed that the embedded spheroids can promote differentiation of stem cells along a preferred lineage or maintain the phenotype of primary cells when they were treated with the correct mixture of growth factors [34].

The present study has some limitations. In particular, it was not possible to provide direct evidence to show whether the PAI-1-induced embedded HPLSC spheroids could promote cementum formation in both in vitro and in vivo conditions. In vitro experiments using a mixture of HPLSC spheroids and dentin are required to evaluate the formation of cementum. We speculate that cementogenic-differentiated cells from HPLSC spheroids may be able to migrate and attach to the surface of dentin within the collagen gel, consequently generating a lamination of cementum on the dentin. Although 3D cell culture aims to better mimic the in vivo situation without the need to maintain an animal facility, it is important to understand whether the embedded HPLSC spheroids that are treated with PAI-1 can promote cementum formation in animal models. Transplantation of the embedded HPLSC spheroids with PAI-1 showed cementogenic differentiation ability in vivo. In vivo transplantation studies have suggested that spheroids that were treated with PAI-1 may generate the fibrous connective tissue of the periodontal ligament containing newly formed cementum-like tissue on the dentin surface. Further in vitro and in vivo studies are warranted to examine the enhancement of cementogenic differentiation using the embedded HPLSC spheroids that are treated with PAI-1 for the potential use in periodontal regeneration.

## 4. Materials and Methods

### 4.1. Cell Culture

The present study used HPLSCs expressing CD146 and STRO-1 that were stored at the Department of Endodontology and Operative Dentistry, Division of Oral Rehabilitation, Faculty of Dental Science, Kyushu University [9]. The cells were cultured in Dulbecco’s Modified Eagle Medium (DMEM; Fuji Film Wako, Japan) that was supplemented with 10% (*v*/*v*) fetal bovine serum (FBS) and 1× Anti-Anti solution (Invitrogen Corporation, Carlsbad, CA, USA) and incubated at 37 °C in an atmosphere of 5% CO_2_. In the 2D monolayer cells experiments, the cells were cultured in an osteogenic induction medium (OIM) that was composed of DMEM supplemented with 10 nM dexamethasone, 200 μM ascorbic acid, 10 mM β-glycerophosphate, and 10% FBS, or OIM containing 100 ng/mL recombinant human plasminogen activator inhibitor 1 (rhPAI-1; PeproTech, Cranberry, NJ, USA).

### 4.2. Collagen-Embedded HPLSC Spheroids

The dissociated HPLSC monolayers were resuspended in DMEM to obtain a single cell suspension. HPLSCs (1 × 10^5^ cells/mL, corresponding to approximately 10,000 cells/well) were added to Corning 96 well Ultralow Attachment microplates (Corning, NY, USA) and were cultured in DMEM for 24 h. The collagen gel mixtures for embedded spheroids were prepared using AteloCell (Koken Co., LTD., Tokyo, Japan) according to the manufacturer’s instructions. The prepared collagen gel mixtures were applied to each well of a 24-well plate, and the collected spheroids were transferred to the mixtures in the plates. The plates were incubated at 37 °C in 5% CO_2_ for 30 min for gelation, and then, DMEM (control), OIM alone (OIM), or OIM containing rhPAI-1 (PAI-1) were added to the wells for 7 and 14 days.

The spheroid size was evaluated by measuring their diameter in the embedded HPLSC spheroids using WinROOF image analysis software. Cell viability in the embedded spheroids was detected using Cultrex 3D Spheroid Cell Proliferation/Viability Assay Kit (Trevigen, MN, USA), according to the manufacturer’s instructions. Absorbance at 570 nm was then measured with the Microplate Reader. 

The motility or migration activity of the constituent cells in the embedded spheroids was determined by the expansion of the spheroids in each group. The area of each spheroid was measured at 24, 48, and 72 h after the collagen-embedding culture was added using the WinROOF image analysis software. We also counted the number of disaggregated spheroids using the WinROOF software.

### 4.3. Western Blot Analysis

The monolayer and spheroid HPLSCs were lysed in cell lysis buffer (Cell Signaling Technology, Danvers, MA, USA) containing 1× Protease/Phosphatase Inhibitor Cocktail (Cell Signaling Technology). The protein concentrations were measured using the Pierce BCA Protein Assay Kit (Thermo Fisher Scientific, Rockford, IL, USA). Equal amounts (15 μg) of protein and protein marker (Precision Plus Protein Western C Standards; Bio-Rad Laboratories, Hercules, CA, USA) were separated on Mini-PROTEAN TGX Precast Gels for 30 min at 200 V. The Trans-Blot Turbo Transfer System (Bio-Rad Laboratories) was used to transfer the separated proteins to a polyvinylidene fluoride membrane. Western blots were processed using iBind Western System (Life Technologies, Carlsbad, CA, USA) with primary antibodies, rabbit anti-CEMP1 antibody (×1000: ab134231, Abeam Inc., Cambridge, UK) and mouse anti-CAP antibody (3G9) (×1000: sc-53947, Santa Cruz Biotechnology Inc., Dallas, Texas, USA), and goat antimouse or antirabbit IgG (H+L)-HRP conjugate (×1000: 1706516, 1706515, Bio-Rad Laboratory, Hercules, CA, USA) as secondary antibodies. The mouse anti-actin beta (β-actin) antibody (×1500: VMA00048, Bio-Rad Laboratories, Hercules, CA, USA) was used as a loading control. An enhanced chemiluminescence system (SignalFire Plus ECL Reagent; Cell Signaling Technology, Danvers, MA, USA) was used to develop the protein bands. The protein levels were quantified by densitometry using an ImageQuant LAS 4000 bimolecular imager (GE Healthcare, Uppsala, Sweden).

### 4.4. Immunocytochemistry Analysis

The monolayered HPLSCs and collagen-embedded HPLSC spheroids that were treated with collagenase were fixed with 4% paraformaldehyde for 10 min and then washed in 0.3% Triton-X in phosphate-buffered saline (PBS) for 10 min. The cells were incubated with rabbit anti-CEMP1 antibody (×1000: ab134231, Abeam Inc., Cambridge, UK) and mouse anti-CAP antibody (3G9) (×1000: sc-53947, Santa Cruz Biotechnology Inc., Dallas, Texas, USA) at 4 °C overnight. The cells were washed with PBS and then incubated with goat antimouse or antirabbit IgG (H+L) secondary antibody-Alexa Fluor Plus 488 or 588 (×400: A32723, A32731, A-11004, A-11011, Thermo Fisher Scientific, Waltham, MA, USA) at room temperature for 45 min. The nuclei were visualized by counterstaining the cells with Hoechst 33342 (5 μg/mL; Sigma-Aldrich Corporation, St. Louis, MO, USA). Using the WinROOF image analysis software, positive and negative cells were automatically counted and the percentage (%) of positive cells was calculated. 

### 4.5. Alkaline Phosphatase (ALP) and Alizarin Red S (ARS) Staining

HPLSCs were stained using an ALP kit (Sigma-Aldrich Corporation) containing Fast Red Violet solution and naphthol AS-BI phosphate solution according to the manufacturer’s instructions. For ARS staining, the cells were stained for 5 min with 1% Alizarin red solution. The excess dye was then removed, and the cells were washed with distilled water. Staining intensities of ALP and ARS were estimated using the WinROOF image analysis software (version: 3.8.0, Mitani Corporation, Fukui, Japan).

### 4.6. Statistical Analysis

All the statistical analyses were performed using EZR (Saitama Medical Sente, Juchi Medical University, Saitama, Japan), which is a graphical user interface for R (The R Foundation for Statistical Computing, Vienna, Austria) that is a modified version of R commander that is designed to add statistical functions that are frequently used in biostatistics. Analyses were performed using one-way analysis of variance with Bonferroni’s multiple comparison test to determine statistical differences among the samples. Data are presented as the mean ± standard deviation (SD) and *p*-values < 0.05 were considered statistically significant.

## 5. Conclusions

The present study revealed that rhPAI-1 treatment of embedded HPLSC spheroids enhanced the cementogenic differentiation via interplay between the spheroids and the collagen in the ECM compared with conventional 2D culture and other conditions in 3D culture. Although there are various approaches to improve efficiency of MSC differentiation, embedded spheroids are a powerful tool to examine cell differentiation and tissue engineering and enable the recapitulation of in vivo environments within an in vitro setting.

## Figures and Tables

**Figure 1 ijms-23-02340-f001:**
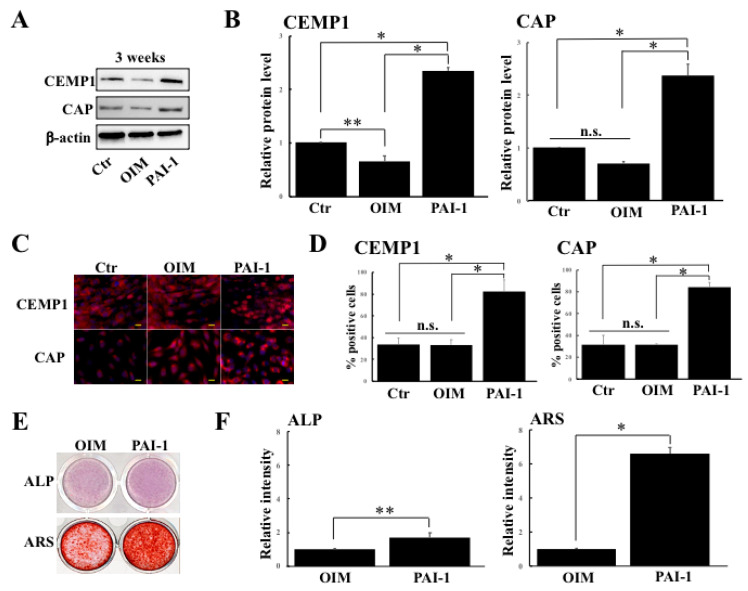
The effect of rhPAI-1 on cementogenic differentiation of HPLSCs in 2D culture. (**A**) Western blot of cementum protein 1 (CEMP1) and cementum attachment protein (CAP) in HPLSCs that were cultured in 2D with DMEM (control), osteogenic inducing medium (OIM) alone, and OIM containing rhPAI-1 (PAI-1; 100 ng/mL) for 3 weeks using β-actin as a loading control. (**B**) Quantification of the protein expression was performed using the ImageQuant LAS4000 biomolecular imager. The relative levels of CEMP1 and CAP expression were normalized to β-actin and were expressed as fold changes compared with HPLSCs that were cultured with DMEM (control). Data represent the mean ± SD of three independent experiments. n.s., not significant, * *p* < 0.01, ** *p* < 0.05. (**C**) Fluorescence images of the HPLSCs that were stained with CEMP1 (red) and CAP (red) in 2D culture with DMEM (control), OIM alone (OIM), and OIM containing rhPAI-1 (100 ng/mL) for 3 weeks. The nuclei were counterstained with DAPI (blue). The scale bar represents 20 μm. (**D**) Semiquantification of the percentage of positive cells that were stained with CEMP1 (nuclei) and CAP (perinuclear cytoplasm). Data represent the mean ± SD of three independent experiments. n.s., not significant, * *p* < 0.001. (**E**) Staining for ALP activity in the HPLSCs in 2D culture with OIM alone (OIM) or with PAI-1 for 3 weeks. Alizarin red-staining (ARS) was also performed in the cells that were cultured with OIM alone or with PAI-1 for 3 weeks. (**F**) Semiquantification of the staining intensities of ALP and ARS using the WinROOF image analysis software. The relative staining intensities of both ALP and ARS of HPLSCs that were cultured with OIM were expressed as fold changes. Data are displayed as the mean ± SD of three independent experiments. * *p* < 0.001, ** *p* < 0.05.

**Figure 2 ijms-23-02340-f002:**
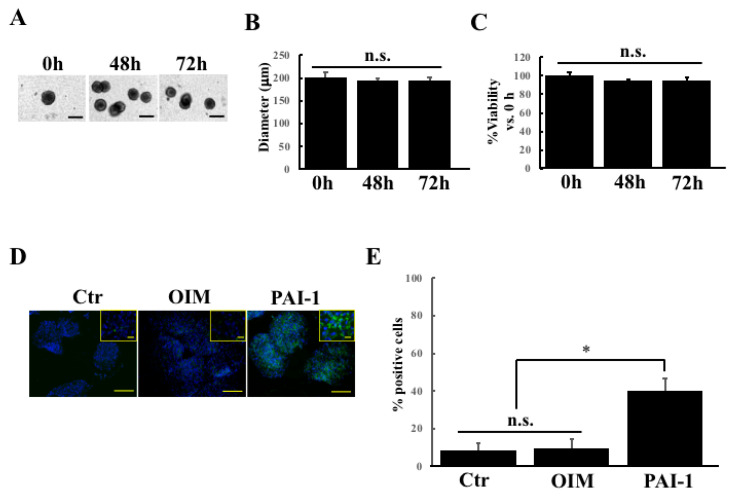
Effect of rhPAI-1 on the CEMP1 expression in the embedded HPLSC spheroids. (**A**) Phase-contrast images of the HPLSC spheroids at 0, 48, and 72 h after the embedded culture with DMEM. The scale bars represent 200 μm. (**B**) Diameter measurements of spheroids at 0, 48, and 72 h after the embedded culture. The analysis was performed using WinROOF software and the data represent the mean ± SD of three independent experiments. n.s., not significant. (**C**) The cell viability of the spheroids was measured at 0, 48, and 72 h after the embedded culture as absorbance in the independent experiments with six replicates each. The results are expressed as a mean value ± SD percent optical density vs. 0 h spheroids. n.s., not significant. (**D**) Immunostaining images of CEMP1 (green) in HPLSC spheroids after a 2 day incubation with DMEM (control), OIM alone (OIM), or OIM containing rhPAI-1 (PAI-1). The nuclei were counterstained with DAPI (blue). The scale bars represent 100 μm. Pictures at high magnification are shown in the insert. The scale bars represent 20 μm. (**E**) Semiquantification of the percentage of positive cells that were stained with CEMP1 in the embedded HPLSC spheroids. The percentage (%) of positive cells was calculated by the number of positive cells/the number of whole cells in the spheroids using the WinROOF software. Data represent the mean ± SD of three independent experiments. * *p* < 0.01.

**Figure 3 ijms-23-02340-f003:**
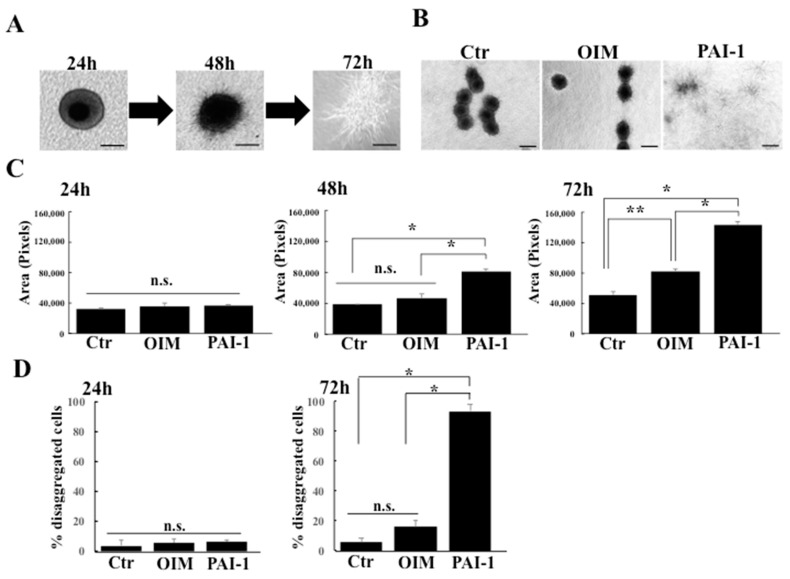
Morphological changes in the embedded spheroid. (**A**) Phase-contrast images of sequential morphological changes in the embedded spheroid cultured in OIM containing rhPAI-1 from 24 h to 72 h. The scale bar represents 100 μm. (**B**) Comparative images of the embedded spheroids that were cultured in DMEM (control), OIM alone (OIM), and OIM containing rhPAI-1 (PAI-1) at 72 h. The scale bar represents 100 μm. (**C**) The extended areas of the embedded spheroids that were cultured in DMEM (Ctr), OIM alone (OIM), and OIM containing rhPAI-1 were measured at 24, 48, and 72 h using WinROOF image analysis software. Data represent the mean ± SD of three independent experiments. n.s., not significant, * *p* < 0.001, ** *p* < 0.05. (**D**) Semiquantification of the percentages of disaggregated spheroids at 24 and 72 h using the WinROOF software. Data represent the mean ± SD of three independent experiments. * *p* < 0.001. n.s., not significant.

**Figure 4 ijms-23-02340-f004:**
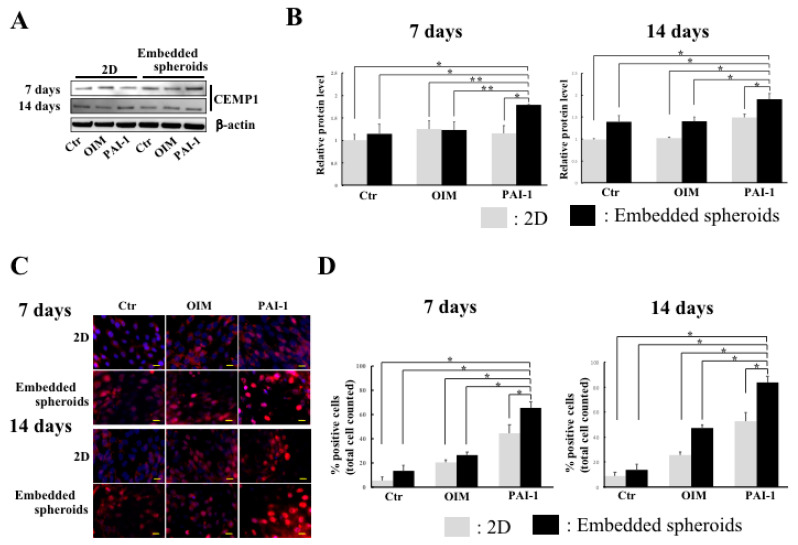
Increased CEMP1 expression in the rhPAI-1-treated embedded HPLSC spheroids. (**A**) Western blot of the CEMP1 expression in 2D-monolayered HPLSCs and the embedded HPLSC spheroids that were cultured in DMEM (control), OIM alone (OIM), and OIM containing rhPAI-1 (PAI-1; 100 ng/mL) for 7 and 14 days. (**B**) Quantification graphs of the protein expression were performed using the ImageQuant LAS4000 biomolecular imager. The relative levels of CEMP1 expression at 7 and 14 days were normalized with β-actin and expressed as fold-changes compared with 2D monolayered HPLSCs that were cultured with DMEM (control). Data represent the mean ± SD of three independent experiments. * *p* < 0.01, ** *p* < 0.05. (**C**) Fluorescence images of CEMP1 staining (red) in 2D-monolayered HPLSCs and the embedded HPLSC spheroids that were cultured in DMEM (control), OIM alone (OIM), and OIM containing rhPAI-1 (PAI-1; 100 ng/mL) up to 14 days. Nuclei were counterstained with DAPI (blue). The scale bar represents 20 μm. (**D**) Semiquantification of the percentage of positive cells that were stained with CEMP1 at 7 and 14 days. Data represent the mean ± SD of three independent experiments. * *p* < 0.01.

## Data Availability

Data are contained within the article.

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
