# Peer review of "Embedded Human Periodontal Ligament Stem Cells Spheroids Enhance Cementogenic Differentiation via Plasminogen Activator Inhibitor 1"

_ijms, 2022, doi:10.3390/ijms23042340_

Round 1
Reviewer 1 Report
The manuscript is lacking some important experiments to validate the main conclusions. The following concerns should be addressed before the manuscript can be considered for publication.
- Line 40 – The authors refer that MSCs are mesenchymal stem/stromal cells. Thus, they can not state that MSCs are multipotent stem cells. They should include “stromal”.
- Line 42 and line 43– The term mesenchymal stem cells should be changed to mesenchymal stromal cells.
- Line 48 – What about chondrogenic differentiation?
- The authors need to include information about the different donors used in the manuscript (age, gender, …).
- The authors should include results regarding HPLSCs characterization, such as flow cytometry analysis and multilineage differentiation assays.
- The authors should include results from more than one donor. Although they have performed the experiments three times, the number of donors used is 1 (n=1).
- Regarding Figure 1, all the results presented are based on qualitative assays. The authors should include quantitative data, such as calcium concentration and ALP activity quantification and osteogenic gene expression analysis.
- How did the authors choose the concentration of PAI-1?
- Why were CEMP1 expression levels low when cells were cultured under osteogenic differentiation medium without PAI-1? Osteogenic differentiation of HPLSCs should enhance CEMP1 expression levels.
- The results presented in the manuscript do not demonstrate the impact of PAI-1 alone. The authors should include a condition in which PAI-1 is added to cells cultured in DMEM medium instead of osteogenic differentiation medium.
- Regarding HPLSC spheroids, why did the authors only analyze CEMP1 expression after 2 days of cell culture?
- The authors claim that PAI-1 treatment can enhance cementogenic differentiation in HPLSC spheroids based on immunocytochemistry analysis. However, more assays should be performed to confirm these results.
- Regarding the interplay of HPLSCs and collagen, the authors observed that spheroids treated with PAI-1 presented higher disaggregation, however the authors need to understand if the cells that migrated from the spheroids are viable. More assays are required.
Author Response
- Line 40 – The authors refer that MSCs are mesenchymal stem/stromal cells. Thus, they can not state that MSCs are multipotent stem cells. They should include “stromal”.
Response:
Following the reviewer’s comment, we revised the text as follows (added word is underlined):
Mesenchymal stem/stromal cells (MSCs) are multipoint somatic stem/stromal cells that can—— (Line 40).
- Line 42 and line 43– The term mesenchymal stem cells should be changed to mesenchymal stromal cells.
Response:
Following the reviewer’s comment, we revised the text as follows (added word is underlined):
The use of dental mesenchymal stem/ stromal cells for periodontal tissue formation (Line 43).
- Line 48 – What about chondrogenic differentiation?
Response:
Thank you for the reviewer’s comment. We revised the text as follows (added word is underlined):
Particularly, HPLSCs have the potential to differentiate into osteoblasts, chondroblasts, and adipocytes. (Line 49)
- The authors need to include information about the different donors used in the manuscript (age, gender, …).
Response:
In this study, the primary cell culture of human periodontal ligament stem cells we used was not from volunteers. We used an established human periodontal ligament stem cell (HPLSC) line [9].
- The authors should include results regarding HPLSCs characterization, such as flow cytometry analysis and multilineage differentiation assays.
Response:
Characterization of the HPLSC cell line used in this study, including flow cytometry analysis and multilineage differentiation assays, has been already reported [9]. Thus, we did not present those in this manuscript and instead added stem cell markers expressed by the HPLSCs used in this study in Section 4.1. Cell culture of the Materials and Methods section as follows (added information is underlined):
The present study used HPLSCs expressing CD146 and STRO-1 that were stored at the Department of Endodontology and Operative Dentistry, Division of Oral Rehabilitation, Faculty of Dental Science, Kyushu University [9]. (Line 346-348)
- The authors should include results from more than one donor. Although they have performed the experiments three times, the number of donors used is 1 (n=1).
Response:
As described above, In this study, the primary cell culture of human periodontal ligament stem cells we used was not from volunteers. We used an established human periodontal ligament stem cell (HPLSC) line [9].
- Regarding Figure 1, all the results presented are based on qualitative assays. The authors should include quantitative data, such as calcium concentration and ALP activity quantification and osteogenic gene expression analysis.
Response:
Following the reviewer’s comment, we have added new data on the staining intensity of ALP and ARS in Figure 1F. We have also revised the text in Section 4.5. Alkaline phosphatase (ALP) and alizarin red S (ARS) staining of the Materials and Methods section, 2.1. rhPAI-1 enhances cementogenic differentiation of HPLSCs in 2D culture of the Results, and the legend of Figure 1 as follows (added and/or revised sentences are underlined):
1) Materials and Methods: 4.5. Alkaline phosphatase (ALP) and alizarin red S (ARS) staining
HPLSCs were stained using an ALP kit (Sigma-Aldrich Corporation) containing Fast Red Violet and naphthol AS-BI phosphate solutions according to the manufacturer’s instructions. For ARS staining, cells were stained for 5 min with 1% Alizarin red solution. The excess dye was then removed, and the cells were washed with distilled water. Staining intensities of ALP and ARS were estimated using the WinROOF image analysis software (version: 3.8.0, Mitani Corporation, Fukui, Japan). (Line 410-416)
2) Results: 2.1. rhPAI-1 enhances cementogenic differentiation of HPLSCs in 2D culture
Next, we examined ALP levels and Alizarin red intensity in HPLSCs treated with OIM or with PAI-1 for 3 weeks (Fig. 1E, F). The ALP staining intensity of HPLSCs increased in PAI-1-treated cells compared with OIM-treated cells (Fig. 1F). Similarly, Alizarin red S staining revealed higher calcium deposition in the PAI-1 group (Fig. 1F). (Line 142-145)
3) Figure legends: Figure 1
(F) Semiquantification of the staining intensities of ALP and ARS using the WinROOF image analysis software. Relative staining intensities of both ALP and ARS of HPLSCs cultured with OIM were expressed as fold changes. Data are displayed as the mean ± SD of three independent experiments. *P < 0.001, **P < 0.05. (Line 119-122)
- How did the authors choose the concentration of PAI-1?
Response:
We chose the concentration of PAI-1 (100 ng/mL) according to our results from preliminary experiments, which were designed along with the cited literature [8] in this manuscript.
- Why were CEMP1 expression levels low when cells were cultured under osteogenic differentiation medium without PAI-1? Osteogenic differentiation of HPLSCs should enhance CEMP1 expression levels.
Response:
Thank you for raising an important point. In our study, both CEMP1 and CAP expression were suppressed under the stimulation with OIM alone. These findings were also unexpected for us. Nevertheless, a recent study also reported the same results that treatment with osteogenic differentiation medium showed a downregulation of CEMP1 and CAP expression in periodontal ligament stem cells. We suggest that the osteogenic stimulus may induce HPLSCs to differentiate only into osteoblasts while suppressing cementogenic pathways. We revised the second paragraph of the Discussion as follows (added and revised sentences are underlined):
The present study revealed that treatment with rhPAI-1 enhanced CEMP1 and CAP expression in 2D monolayer HPLSCs, whereas the stimulation with OIM alone suppressed the expression of both. A recent study also reported that osteogenic differentiation medium downregulated CEMP1 and CAP expression in periodontal ligament stem cells [10]. These findings may indicate that the osteogenic stimulus may induce HPLSCs to differentiate only into osteoblasts while suppressing cementogenic pathways. Our results that showed upregulation of CEMP1 and CAP by treatment with PAI-1 support the findings of other reports that examined the effects of PAI-1 on induced cementogeniesis in human periodontal ligament stem cells. Several studies have reported that CEMP1 protein expression is localized to the cementoblast cell layer lining of the cementogenesis surface of cementum [15, 32]. Monolayer HPLSCs in the PAI-1-treated group were more susceptible to cementogenic differentiation than HPLSCs in the other groups. Furthermore, CEMP1 may play an important role in cementogenesis in response to rhPAI-1. A previous study examining the role of PAI-1 in osteogenesis revealed that protein levels of the osteoblastic-specific factors, Runx2 and OSX, were increased by rhPAI-1 treatment [33]. Although it remains unclear which transcription factors may be crucial for cementogenic differentiation induced by rhPAI-1, our findings suggest that rhPAI-1 promotes cementoblastic differentiation in 2D monolayer HPLSCs. (Line 270-287)
- The results presented in the manuscript do not demonstrate the impact of PAI-1 alone. The authors should include a condition in which PAI-1 is added to cells cultured in DMEM medium instead of osteogenic differentiation medium.
Response:
Following the reviewer’s comments, we added the results of the cells treated with PAI-1 alone in Section 2.1. rhPAI-1 enhances cementogenic differentiation of HPLSCs in 2D culture of the Results as follows (added sentences are underlined):
2.1. rhPAI-1 enhances cementogenic differentiation of HPLSCs in 2D culture
Since PAI-1 is known to stimulate cementogenesis, we first examined whether PAI-1 promoted cementogenic differentiation in HPLSCs in 2D culture. HPLSCs were cultured in DMEM alone (control) or in OIM without or with rhPAI-1 (100 ng/mL) for 3 weeks as previously described [8]. Protein levels of the cementogenic markers were determined using western blot analysis (Fig. 1A, B). After 3 weeks, the expression levels of CEMP1 and CAP in the HPLSCs were significantly higher in the PAI-1-treated cells than in the control (P < 0.01) and in the cells in OIM alone (P < 0.01) (Fig. 1B). In the cells treated with PAI-1 alone, the expression levels of both CEMP1 and CAP were identical to those in the control (data not shown). Immunocytochemistry analysis revealed that CEMP1 is expressed in the nuclei of HPLSCs treated with PAI-1 (Fig. 1C). In the immunocytochemical staining of CAP, HPLSCs treated with PAI-1 fluoresced in the perinuclear area (Fig. 1C). The percentage of CEMP1-positive cells was significantly different between the PAI-1 (82.6% ± 10.7%) and the control (33.8% ± 5.8%; P < 0.001) and OIM (33.3% ± 5.0%; P < 0.05) groups (Fig. 1D). In CAP staining, the percentage of positive cells was significantly higher in the PAI-1-treated HPLSCs (84.4% ± 4.4%) than in the control (31.7% ± 8.7%; P < 0.001) or the HPLSCs treated with OIM alone (31.5% ± 0.5%; P < 0.001) (Fig. 1D). Immunocytochemistry results of the cells treated with PAI-1 alone were similar to those of the control (data not shown). Next, we examined the ALP levels and Alizarin red intensity in HPLSCs treated with OIM alone or with PAI-1 for 3 weeks (Figs. 1E, F). The ALP staining intensity of HPLSCs increased more in the PAI-1-treated cells than in the OIM-treated cells (Fig. 1F). Similarly, Alizarin red S staining revealed higher calcium deposition in the PAI-1 group (Fig. 1F). These findings revealed that rhPAI-1 could induce differentiation of HPLSCs into cementoblasts as mineral-forming cells in 2D culture. (Line 125-147)
- Regarding HPLSC spheroids, why did the authors only analyze CEMP1 expression after 2 days of cell culture?
Response:
To examine CEMP1 expression in the constituent cells of the spheroid body, we performed an immunocytochemical assay in the HPLSC spheroids at day 2 of the embedded culture. After 3 days of culture, embedded spheroids treated with PAI-1 became the starting point of spheroid disaggregation.
- The authors claim that PAI-1 treatment can enhance cementogenic differentiation in HPLSC spheroids based on immunocytochemistry analysis. However, more assays should be performed to confirm these results.
Response:
Although we unfortunately could not show the results of CEMP1 expression in HPLSC spheroids at day 2 using western blot, we exhibited the results of both immunocytochemistry and western blot analyses in HPLSC spheroids after 7 and 14 days of collagen-embedded culture in Figure 4. We believe that these illustrate the effects of PAI-1 on the enhancement of cementogenesis in HPLSC spheroids.
- Regarding the interplay of HPLSCs and collagen, the authors observed that spheroids treated with PAI-1 presented higher disaggregation, however the authors need to understand if the cells that migrated from the spheroids are viable. More assays are required.
Response:
Following the reviewer’s comment, we added new data on the “mortality or migration activity of constituent cells in embedded spheroids” in Figure 3C. We revised the text in the Results, Materials and Methods section, and legend of Figure 3 as follows (added and/or revised sentences are underlined):
1) Results “2.3. PAI-1 promotes interplay between HPLSC spheroids and collagenous ECM”
We examined the morphological changes in embedded spheroids cultured in DMEM (control), OIM alone (OIM), and OIM containing rhPIA-1 (PIA-1) to elucidate the dynamics of HPSLC spheroids within collagen gel. Ball-like shapes were obtained in all HPLSC spheroids embedded in collagen gel at 24 h, regardless of treatment. The surface of embedded spheroids treated with PAI-1 was covered with many spines at 48 h, showing a sea urchin-like morphology (Fig. 3A), whereas other groups (control and OIM) remained as ball-like spheroids. These results indicate that PAI-1 can promote cell growth from the embedded spheroids. After 72 h of PAI-1 treatment, many spheroids were disaggregated, and isolated cells had migrated and spread into the collagen gel (Fig. 3A). By contrast, spheroids in the control and OIM groups remained as ball-like or sea urchin-like shapes at 72 h (Fig. 3B). To test the potential effects of PAI-1 on the motility or migration activity of the constituent cells in embedded spheroids to collagen gels, we measured the extended area of the spheroids at 24, 48, and 72 h after embedded culture. There were no significant differences between the spheroid ranges of the groups at 24 h (Fig. 3C). After 48 h, the expanded area of spheroids treated with PAI-1 was greater than the control and OIM-treated groups (Fig. 3C). Together, these results (Fig. 3B, C) demonstrate a key role for PAI-1 in promoting cell migration to the ECM in HPLSC spheroids. During cell migration from the embedded spheroids, the spheroids disaggregate (Fig. 3A, B). Next, we examined the effect of PAI-1 treatment on the disaggregation of the spheroids by calculating the percentage of disaggregated spheroids using the WinROOF image analysis software. At 24 h, there was no significant difference between the disaggregation percentages of the three groups (control vs. PAI-1, P = 0.608; OIM vs. PAI-1, P = 0.966) (Fig. 3D). The PIA-1 group showed a higher percentage of embedded spheroid disaggregation (93.7% ± 4.2%) at 72 h compared with the control (5.8% ± 2.3%) and OIM (16.0% ± 4.6%) groups (P < 0.001 for both) (Fig. 3D). These findings suggest that PAI-1 promotes interplay between spheroids and collagen, resulting in the disaggregation of spheroids via cell migration. (Line 194-220)
2) Materials and Methods “4.2. Preparation of collagen-embedded HPLSC spheroids”
Dissociated HPLSC monolayers were resuspended in DMEM to obtain a single cell suspension. HPLSCs (1 × 105 cells/mL, corresponding to approximately 10,000 cells/well) were added to Corning 96 well Ultralow Attachment microplates (Corning, NY, USA) and were cultured in DMEM for 24 h. Collagen gel mixtures for embedded spheroids were prepared using AteloCell (Koken Co., LTD., Tokyo, Japan) according to the manufacturer’s instructions. The prepared collagen gel mixtures were applied to each well of a 24-well plate, and the collected spheroids were transferred to the mixtures in the plates. The plates were incubated at 37℃ in 5% CO2 for 30 min for gelation, and then, DMEM (control), OIM alone (OIM), or OIM containing rhPAI-1 (PAI-1) were added to the wells for 7 and 14 days.
Spheroid size was evaluated by measuring their diameter in the embedded HPLSC spheroids using WinROOF image analysis software. Cell viability in the embedded spheroids was detected using Cultrex 3D Spheroid Cell Proliferation/Viability Assay Kit (Trevigen, MN, USA), according to the manufacturer’s instructions. Absorbance at 570 nm was then measured with the Microplate Reader.
The motility or migration activity of constituent cells in embedded spheroids was determined by the expansion of the spheroids in each group. The area of each spheroid was measured at 24, 48, and 72 h after the collagen embedding culture was added using the WinROOF image analysis software. We also counted the number of disaggregated spheroids using the WinROOF software. (Line 356-376)
3) Legend of “Figure 3”
Figure 3. Morphological changes in the embedded spheroids. (A) Phase-contrast images of sequential morphological changes in the embedded spheroid cultured in OIM containing rhPAI-1 from 24 to 72 h. Scale bar represents 100 µm. (B) Comparative images of embedded spheroids cultured in DMEM (control), OIM alone (OIM), and OIM containing rhPAI-1 (PAI-1) at 72 h. Scale bar represents 200 µm. (C) The extended areas of the embedded spheroids cultured in DMEM (Ctr), OIM alone (OIM), and OIM containing rhPAI-1 were measured at 24, 48, and 72 h using WinROOF image analysis software. Data represent the mean ± SD of three independent experiments. n.s., not significant, *P < 0.001, **P < 0.05. (D) Semiquantification of the percentages of disaggregated spheroids at 24 and 72 h using the WinROOF software. Data represent the mean ± SD of three independent experiments. *P < 0.001. n.s., not significant. (Line 184-193)

Reviewer 2 Report
All my comments were seriously addressed, by also performing suggested and requested experiments. In general, after the revision this paper has been improved. Great job authors, thanks and compliments
Author Response
All my comments were seriously addressed, by also performing suggested and requested experiments. In general, after the revision this paper has been improved. Great job authors, thanks and compliments.
Response:
We appreciate the reviewer’s cooperation.
Round 2
Reviewer 1 Report
The authors have addressed most the reviewer's comments.
This manuscript is a resubmission of an earlier submission. The following is a list of the peer review reports and author responses from that submission.
Round 1
Reviewer 1 Report
The manuscript is lacking some important experiments to validate the main conclusions. The following concerns should be addressed before the manuscript can be considered for publication.
Introduction:
- The authors claim that cementum formation is critical for the development of periodontium. Please, describe the importance of cementum (line 33).
- Please, include which stromal cell markers are expressed in human PLSCs (line 42).
- The authors compare HPLSCs to “adult MSC”. However, they need to explain which adult MSC are they referring to (line 42).
- The role of PAI-1 in cementogenesis should be better described.
- The authors need to explain what is a multicellular spheroid (line 78).
Materials and Methods:
- The authors need to include information about the different donors used in the manuscript (age, gender, …).
- Please, include information regarding the concentration of Alizarin Red solution used in the experiments.
Results:
- The authors should include results regarding HPLSCs characterization, such as flow cytometry analysis and multilineage differentiation assays.
- The authors should include results from more than one donor. Although they have performed the experiments three times, the number of donors used is 1 (n=1).
- Regarding Figure 1, all the results presented are based on qualitative assays. The authors should include quantitative data, such as calcium concentration and ALP activity quantification and osteogenic gene expression analysis.
- How did the authors quantify the % of CEMP1 cells based on immunocytochemistry images?
- How did the authors choose the concentration of PAI-1?
- Why were CEMP1 expression levels low when cells were cultured under osteogenic differentiation medium without PAI-1? Osteogenic differentiation of HPLSCs should enhance CEMP1 expression levels.
- Line 99- Please correct “Sine”.
- The results presented in the manuscript do not demonstrate the impact of PAI-1 alone. The authors should include a condition in which PAI-1 is added to cells cultured in DMEM medium instead of osteogenic differentiation medium.
- Line 103 – The authors should rewrite this sentence. It is not described which condition showed upregulation of CEMP1 protein levels.
- Line 111- 113- The authors claim that the ALP staining intensity of HPLSCs increased in PAI-1-treated cells compared with OIM-treated cells, however this can not be observed in Figure 1. Quantitative assays should be performed to confirm that PAI-1 can promote cementogenic differentiation of HPLSCs.
- Regarding HPLSC spheroids, why did the authors only analyze CEMP1 expression after 2 days of cell culture?
- The authors claim that PAI-1 treatment can enhance cementogenic differentiation in HPLSC spheroids based on immunocytochemistry analysis. However, more assays should be performed to confirm these results.
- Regarding the interplay of HPLSCs and collagen, the authors observed that spheroids treated with PAI-1 presented higher disaggregation, however the authors need to understand if the cells that migrated from the spheroids are viable. More assays are required.
- Regarding Figure 4, why did the authors perform the experiment for 7 days? The authors need to include more quantitative data.
- Regarding Figure 4D, are these results normalized to the total number of cells?
Reviewer 2 Report
The manuscript entitled "Embedded human periodontal ligament stem cells spheroids enhance cementogenic differentiation via plasminogen activator inhibitor 1" faces a very interesting topic and is well written.
Line 99: change Sine with Since.
The literature is old and needs to be updated.
Reviewer 3 Report
Broad comments
The research is interesting and the message can advantage the development of oral regenerative medicine strategies and attract readership of experts in mesenchymal stem cells field. However, a revision is highly demanded for making this paper worthy of the journal selected for the submission.
Major issues
1) At least one other protein osteogenic marker should be analysed. Do you have some remaining protein lysate and consider osteocalcin or osteopontin or sialoprotein or RunX2 or other cementogenic renowned marker. Please have a look and be helped by following paper 10.1007/s00441-016-2513-8 (suggested to cite).
2) Figure 2B must be changed, removing the current green background in both “Ctr” and “OIM” pictures that makes impossible to compare CEMP right fluorescence with “PAI-1” picture. Thanks. Alternatively new picture without background (it should be black) can be used and inserted with high resolution.
3) The sentence in the legend of Figure 2 “Phase-contrast image of embedded HPLSC spheroids for 2 days after embedding of 25 spheroids into collagen gel.” (at lines 116-117). Please revise, something is missing or sentence is not clear.
4) The dimensions of the embedded spheroids at 2 days is not clear. The scale bar of figure 2A says that are almost 20 um and conversely the scale bar representing 100 uM is minor than the dimension of the diameter of the spheroid (guessing is about 200 um). Please clarify and correct the information. Scale bar typo? Different spheroids?
5) Quantification of Figure 2C should be explained in details in supplementary info. In 4.4. Authors reported “The percentage of positive cells was estimated under a 40× objective lens of a fluorescence microscopy at five different sites for each specimen.” , however, I cannot understand the threshold (Manually? Subjective? Arbitrary) of the fluorescence for the counted single cells.
6) Scale bars are not present on Figure 3°. Please add to the relative legend or overlay on the picture.
Again the spheroids seem have a new dimension also in Ctr that did not show extensive migration and disaggregation. However the spheroids are visualized at 72h instead of 48h from embedding. Thus, may the authors write (if I well understood) that the dimension change because the spheroid compacted with prolonged time of embedding culture?
7) Although is written as semi-quantification by the authors, the number of disaggregated cells only by phase contrast is not very acceptable. Can the authors try to use some software tools or considering the area and the black intensity peaks per area in order to better show their results? A comparison between pictures of ll the 3 experimental conditions CTRL, OIM and PAI-1 at 24h (all ball-like and not currently added to the panel B of the Figure) and the ones at 72h (currently shown in panel B)
8) Figure 4C pictures should have higher quality.
9) I can observe that 2D-cultured cells are low-density differentiated samples (Fig- 4C), how is it possible since osteogenic differentiation is know to promote also cell growth at least in OIM?
10) Why is CEMP analyzed only after 7 days of osteogenic induction? Please specify it and the limitation of this analysis to the readers.
11) Protein expression of CEMP by Western Blot at 2 or 3 weeks can confirm the results of 7 days and should be added to the research study, since protein expression was analyzed too early (only 7 days) and can be transient and ineffective at translational level or in vivo. Alternatively gene expression of CEMP at 1 and 2 weeks (2 time-points between 7 and 15 days) of differentiation I will support authors’ request of additional time for completing any revision, in order to perform such experiments. Please kindly ask to editors, thanks.
12) No viability of the HPLSC spheroid were added. Any live&dead staining or MTT/Resazurin/ATP-based assessment can be included int he manuscript, even without OIM e.g. just in Ctr spheroids before and after 72h of embedding? 217-221 lines also remarks possibility of variable necrotic cores.
13) After 66th line, the authors should mention that MSC spheroids may have great relevant secretory activity, but heterogeneity of different sources of MSCs should be considered and techniques demanded to measure the discrepancy (please have a look and find appropriate reference at doi.org/10.3390/antibiotics10070750
14) I suggest to summarize the content of a previous study (doi.org/10.1109/TNB.2020.2984551) including osteogenic differentiation of HPLSCs treated with conditioned medium of cell transfected with PAI-1 and cite in the introduction or in discussion of the manuscript
15) Similar to the previous point, please cite the most recent findings similar to this manuscript aim. Please have a look at doi.org/10.1186/s13287-021-02581-6
16) The catalog number for all used antibodies (especially the primary) or at least host, producer and dilution should be included in 4.3 and 4.4 material and methods sections.
Minor comments
17) Typo at line 295 , 15 g (suffix missed?)
18) Typo at line 313, 5 g/mL (suffix missed?)
19) About section 4.2 , the number of days of differentiation (induction by OIM or OIM+test) should be indicated for all the experiments reported (refer to results).
20) Line 216, “prevalent” where?
21) One advice is that Figure 4D y axis should include % = total cell counted, because the cell density showed in the pictures is really different between 2D and 3D